# Transthyretin Is Commonly Upregulated in the Hippocampus of Two Stress-Induced Depression Mouse Models

**DOI:** 10.3390/ijms24043736

**Published:** 2023-02-13

**Authors:** Hidehito Saito-Takatsuji, Yasuo Yoshitomi, Ryo Yamamoto, Takafumi Furuyama, Yasuhito Ishigaki, Nobuo Kato, Hideto Yonekura, Takayuki Ikeda

**Affiliations:** 1Department of Biochemistry, Kanazawa Medical University School of Medicine, 1-1 Daigaku, Uchinada, Kahoku-gun, Ishikawa 920-0293, Japan; 2Department of Physiology, Kanazawa Medical University School of Medicine, 1-1 Daigaku, Uchinada, Kahoku-gun, Ishikawa 920-0293, Japan; 3Division of Molecular and Cell Biology, Medical Research Institute, Kanazawa Medical University, 1-1 Daigaku, Uchinada, Kahoku-gun, Ishikawa 920-0293, Japan

**Keywords:** mouse model of depression, forced swim stress, repeated social defeat stress, hippocampus, gene expression, transthyretin

## Abstract

Chronic stress can affect gene expression in the hippocampus, which alters neural and cerebrovascular functions, thereby contributing to the development of mental disorders such as depression. Although several differentially expressed genes in the depressed brain have been reported, gene expression changes in the stressed brain remain underexplored. Therefore, this study examines hippocampal gene expression in two mouse models of depression induced by forced swim stress (FSS) and repeated social defeat stress (R-SDS). Transthyretin (Ttr) was commonly upregulated in the hippocampus of both mouse models, as determined by microarray, RT-qPCR, and Western blot analyses. Evaluation of the effects of overexpressed Ttr in the hippocampus using adeno-associated virus-mediated gene transfer revealed that TTR overexpression induced depression-like behavior and upregulation of Lcn2 and several proinflammatory genes (Icam1 and Vcam1) in the hippocampus. Upregulation of these inflammation-related genes was confirmed in the hippocampus obtained from mice vulnerable to R-SDS. These results suggest that chronic stress upregulates Ttr expression in the hippocampus and that Ttr upregulation may be involved in the induction of depression-like behavior.

## 1. Introduction

The brain can respond and adapt to various stresses, but prolonged or repeated stress adversely affects brain functions and contributes to the development of mental disorders such as depression [1,2,3]. Depression is one of the mood disorders characterized by mental (e.g., persistently low mood and decreased motivation) and physical symptoms (e.g., insomnia, loss of appetite, and easy tiredness). The global prevalence of depressive disorder, including both persistent and dysthymia cases, is estimated to be approximately 12% [4]. In recent years, several studies have reported that the development of stress-induced depression is significantly associated with impaired neuronal plasticity [5,6,7] and inflammatory responses [8,9]. Cumulative meta-analyses revealed that the levels of proteins involved in inflammatory regulation, such as interleukin-6 and C-reactive proteins, were higher in patients with major depression than in nondepressed controls [10,11], and some neurotrophic factors, such as brain-derived neurotrophic factor (BDNF), were lower in the serum of a patient with depression than in healthy controls [12]. Such studies searching for biomarkers or genes related to depression have the potential to contribute to the development of effective diagnostic and therapeutic tools in the future.

One of the brain regions involved in both memory function and stress responses is the hippocampus. Chronic stress can affect gene expression in the hippocampus, which then alters neural and cerebrovascular functions, resulting in the development of mental disorders such as depression [3]. For example, hippocampal sirtuin 1 (SIRT1) was reduced by loading chronic ultra-mild stress and repeated restraint stress, and pharmacologic and genetic inhibition of hippocampal SIRT1 function increased depression-like behaviors [13]. Furthermore, the neuropeptide VGF, which is robustly regulated by BDNF/TrkB signaling, is downregulated in the dorsal hippocampus of depressed human subjects and mice with chronic social defeat stress [14]. Chronic stress can also affect the expression of genes involved in the vascular system, injury response, and inflammation in the hippocampus [15,16]. In addition to chronic stress, acute stress can induce protein phosphorylation and gene transcription in the hippocampus [17]. Several animal models for human depression have been used to identify gene expression patterns [18]. However, comparisons of gene expressions between different depression models or the identification of differentially expressed genes in multiple depression models have been scarcely explored, whereas previous studies have mainly focused on a single model in this context.

In this study, we prepared two types of depression mouse models, forced swim stress (FSS)-induced and repeated social defeat stress (R-SDS)-induced depression mouse model, and examined gene expression in the hippocampus of the two models. Transthyretin (Ttr) was identified as a gene that was commonly upregulated in the hippocampus of the studied models. We also examined the effects of TTR overexpression in the hippocampus and found that TTR upregulation affected depression-like behavior and the hippocampal expression of several proinflammatory genes.

## 2. Results

### 2.1. Preparation of an FSS-Induced Depression Mouse Model and Identification of Differentially Expressed Genes in the Hippocampus

First, we prepared an FSS-induced depression mouse model (FSS mice) [19,20]. Antidepressant administration improves the inactive state of FSS mice, making them a recognized mouse model of depression [21,22]. Nine-week-old male B6 mice were subjected to FSS (11 min per day for 5 days), after which their depression-like behavior was evaluated (Figure 1A). The moving distance of the FSS mice in the cylinder was decreased daily (Figure 1B). The immobile time in the tail suspension test (TST) on Day 6 was significantly longer in FSS mice (191.85 ± 14.25 s) than in the control mice (146.05 ± 10.56 s) (Figure 1C). Thus, FSS mice were successfully established.

We utilized DNA microarray to identify genes whose expression in the hippocampus was altered in FSS mice. We identified 8 upregulated genes with Log_2_FC > 0.5 and *p* < 0.05 and 14 downregulated genes with Log_2_FC < −0.5 and *p* < 0.05 (Figure 1D). Among the eight upregulated genes, we chose *Ttr* since it was one of the genes with high fold-change. When we confirmed its upregulation by RT-qPCR and Western blotting, the levels of *Ttr* mRNA (2.93-fold) (Figure 1E) and TTR protein (1.82-fold) (Figure 1F) were significantly increased in the hippocampus of the FSS mice compared with the control mice, suggesting that Ttr is upregulated in the hippocampus under depressive stress induced by FSS.

### 2.2. Preparation of an R-SDS-Induced Depression Mouse Model and Identification of Differentially Expressed Genes in the Hippocampus

Next, we prepared another mouse model, an R-SDS-induced depression mouse model (R-SDS mice). Nine-week-old male B6 mice were subjected to R-SDS for 10 days, as described in Materials and Methods, after which their depression-like behavior was evaluated (Figure 2A). In the social interaction test (SIT), we compared the time that the R-SDS and control mice spent in the interaction zone or the avoidance zone with [ICR(+)] or without [ICR(−)] a social target ICR mouse (Figure 2B) and calculated the “avoidance”.

Zone score” or the “interaction zone score”, which is the ratio of the time spent in the avoidance or interaction zone in the presence of an ICR mouse versus in its absence of an ICR mouse [ICR(+)/ICR(−) ratio]. As shown in Figure 2C, most control mice spent more time in the interaction zone than in the avoidance zone in both the presence and absence of an ICR mouse. In contrast, R-SDS mice, not all, spent more time in the avoidance zone than in the interaction zone in the presence of an ICR mouse, as shown in the “susceptible” of R-SDS mice (Figure 2C). The avoidance zone score [ICR(+)/ICR(−) ratio] of the whole R-SDS mice group (2.67 ± 0.69) strikingly exceeded that of the control mice group (0.88 ± 0.11) (Figure 2D). Because the scores of some R-SDS mice did not differ from those of the control mice, we classified the R-SDS mice with avoidance zone scores exceeding 1 as “susceptible” and the others as “resilient” (Figure 2C,D) [23]. Among the R-SDS mice, the avoidance zone score of the susceptible R-SDS mice (3.57 ± 0.81) was significantly higher than that of both the control (0.88 ± 0.11) and resilient R-SDS mice (0.42 ± 0.16) (Figure 2E). Inversely, the interaction zone score of the susceptible R-SDS mice (0.40 ± 0.10) was significantly lower than that of both the control (1.19 ± 0.05) and resilient R-SDS mice (1.15 ± 0.24) (Figure 2F). Total spending time with an ICR mouse in the avoidance zone was significantly longer in the susceptible R-SDS mice (52.10 ± 8.97 s) than in both the control (16.70 ± 1.75 s) and resilient R-SDS mice (9.50 ± 3.37 s) (Figure 2G). In contrast, total spending time with an ICR mouse in the interaction zone was significantly shorter in the susceptible R-SDS mice (22.95 ± 6.33 s) than in both the control (58.34 ± 3.82 s) and resilient R-SDS mice (63.0 ± 17.28 s) (Figure 2H). Overall, these results suggest that susceptible R-SDS mice have a strong aversion to interacting with the ICR mouse. In the sucrose preference test (SPT), the susceptible R-SDS mice (62.93 ± 3.46%) had a significantly lower sucrose preference than the control (78.08 ± 3.51%) and resilient R-SDS mice (90.55 ± 3.63%) (Figure 2I). These results corroborated a previous report [24], indicating that R-SDS-loaded susceptible mice were in a depressive state. Thus, the second depression mouse model was successfully prepared by loading R-SDS.

We performed DNA microarray analyses to identify genes whose expression in the hippocampus of susceptible R-SDS mice differed from that of the control mice. We identified 62 upregulated genes and 95 downregulated genes in the susceptible R-SDS mice (Figure 2J). Among them, *Ttr* was also highly upregulated in the susceptible R-SDS mice (Figure 2J), similar to that in the FSS mice (Figure 1D). To verify the microarray data, we examined the mRNA and protein levels of Ttr in the hippocampus using RT-qPCR and Western blotting, respectively. The mRNA level of *Ttr* in the hippocampus of susceptible R-SDS mice was significantly increased (3.73-fold) compared with the control mice, but no significant difference was observed between the control and resilient R-SDS mice (Figure 2K). The TTR protein level in the hippocampus of susceptible R-SDS mice was also significantly increased (2.62-fold) compared with the control mice (Figure 2L). Overall, these findings indicate that Ttr expression is commonly elevated in the hippocampus of two stress-induced depression mouse models, FSS and R-SDS.

### 2.3. TTR Overexpression in the Mouse Hippocampus Induces Depression-Like Behavior and Upregulation of Proinflammatory Genes

To examine the effects of TTR upregulation in the hippocampus, we bilaterally overexpressed TTR or ZsGreen1 as a control in the hippocampus of male B6 mice using adeno-associated virus (AAV)-mediated gene transfer system (Appendix A). We investigated the depression-like behavior of mice overexpressing TTR (TTR mice) or ZsGreen1 (ZsG mice) by TST at 31 days and SPT at 32 days after the AAV injection (Figure 3A). We found that the immobile time in TST was significantly increased in the TTR mice (223.16 ± 9.6 s) compared to the ZsG mice (182.41 ± 12.97 s) (Figure 3B) and that sucrose preference was significantly decreased in the TTR mice (85.41 ± 2.67%) compared to the ZsG mice (92.39 ± 0.74%) (Figure 3C). These results indicate that TTR mice exhibit depression-like behavior. Furthermore, 35 days after the AAV injection, the TTR protein level was significantly increased in the hippocampus of the TTR mice (4.95-fold) compared to the ZsG mice (Figure 3D), confirming the TTR overexpression using AAV-mediated gene transfer. These results indicate that excess amounts of TTR protein in the hippocampus cause depression-like behavior in mice.

To investigate whether TTR overexpression affected hippocampal gene expression, we focused on the expression of lipocalin 2 (Lcn2), a modulator of inflammation, that showed the highest level of upregulation in the susceptible R-SDS mice (Figure 2J) and that can increase in the hippocampus of mice with chronic social stress [15]. We also focused on the expression of proinflammatory genes because chronic stress can affect the expression of genes involved in inflammation [15,16], and the development of stress-induced depression is significantly associated with inflammatory responses [8,9]. We found that the LCN2 protein level was significantly upregulated in the hippocampus of the TTR mice (1.42-fold) compared to the ZsG mice (Figure 3E). Moreover, the protein levels of intercellular adhesion molecule 1 (ICAM1) (2.36-fold) (Figure 3F) and vascular cell adhesion molecule 1 (VCAM1) (1.53-fold) (Figure 3G) were significantly upregulated in the hippocampus of the TTR mice as compared to the ZsG mice. 

We then investigated the expression of these genes in the hippocampus of susceptible and resilient R-SDS mice as well as in the control mice. The mRNA levels of *Lcn2* (Figure 3H), *Icam1* (Figure 3I), and *Vcam1* (Figure 3J) were significantly upregulated in the hippocampus of the susceptible R-SDS mice compared with those of the control and resilient R-SDS mice. The mRNA levels of these genes tended to increase in the resilient R-SDS mice when compared with those of the control mice, but no significant difference was observed (Figure 3H–J). These results suggest that the resilient R-SDS mice did not exhibit depression-like behavior because chronic stress did not induce Ttr or inflammation-related gene expression. Overall, these findings indicate that chronic stress-induced Ttr upregulation in the hippocampus and then excess TTR protein stimulated the expression of Lcn2 and proinflammatory genes such as Icam1 and Vcam1, resulting in the development of depression-like behavior in the susceptible R-SDS depression mouse model.

## 3. Discussion

In this study, we prepared two depression mouse models, FSS mice (Figure 1A–C) and R-SDS mice (Figure 2A–I), and examined the gene expression changes in the hippocampus of both models. Through DNA microarray analyses, RT-qPCR, and Western blotting, we found that Ttr expression commonly increased in the two models studied (Figure 1D–F and Figure 2J–L), but no other genes were found to be commonly upregulated in the two models studied. 

Although the FSS mice are known as a mouse model of depression that causes a chronic state of behavioral despair [19,20], it is also thought that immobility in the FSS is not depression-like behavior but a coping style, strategy, or learned response [25,26]. Serchov et al. [22] used TST and SPT to demonstrate depression-like behavior in FSS mice, which is consistent with our findings. Furthermore, genetic studies found that homer scaffold protein 1 (Homer1) was reduced in FSS mice [20] and that Homer1 knockout mice exhibited depression-like behavior [27], implying that FSS mice have some, if not all, of the depressive status. In the present study, the Ttr gene was found to be increased in both FSS and R-SDS mouse models, and Ttr overexpression induced depression-like behavior as measured by two different tests, the TST and PST. However, it is important to assess how well mouse models match human depression because some mouse models showed no relation to human depression in central nervous system gene expression changes [18].

TTR is a homotetrameric protein that helps transport thyroid hormones and retinol throughout the body. While it is synthesized mainly in the liver, retinal pigment epithelium, and pancreas, it is also produced in the choroid plexus (CP) and secreted into the cerebrospinal fluid (CSF) [28,29]. Sullivan et al. showed a decrease in TTR protein in the CSF of depression patients [30]. Turner et al., on the other hand, reported that Ttr gene expression was not altered in the CP of depression patients [29]. Thus, Ttr expression in CP and TTR protein concentration in CSF in depression remain debatable. Because Ttr in CSF is unlikely to be transferred into the hippocampus, it is possible that the Ttr protein produced in the hippocampus itself has an impact on hippocampus functions in an autocrine/paracrine manner. Li et al. recently reported its expression in primary and CP-free cultured embryonic hippocampal neurons [31]. Ttr expression was increased by stress in the hippocampus, according to RNA seq analysis of different brain regions from chronic social defeat stressed mice, and Ttr expression was higher in mice susceptible to stress than that in mice resilient to stress [32]. The present study also demonstrated Ttr production in the mouse hippocampus (Figure 1F and Figure 2L). Martinho et al. reported that the rat *Ttr* gene has a glucocorticoid-responsive element in the first intron, and that the psychosocial stress and a stress hormone, glucocorticoid, induced the Ttr expression in rat CP [33]. Furthermore, it was reported that chronic social defeat stress-induced hypercortisolemia in the stress susceptible mice and that glucocorticoid receptor was expressed in the hippocampus of the mice [34]. These findings suggest that the chronic stress-induced Ttr expression in the hippocampus is mediated by the stress hormone. Although the glucocorticoid concentrations were not measured in this study, it is possible that the Ttr expression is regulated by the stress hormone in the depression mouse models used in this study.

The aggregation of misfolded proteins in mutant and wild-type TTR can result in amyloid deposition, which can impair organ function. The clinical syndromes associated with TTR aggregation are familial amyloid polyneuropathy and cardiomyopathy, in which mutant TTR protein aggregates in peripheral and autonomic nerves and heart, respectively, and senile systemic amyloidosis, in which wild-type protein deposits primarily in the heart and gut [35].

To examine the effects of elevated Ttr expression in the hippocampus, we overexpressed TTR in the hippocampus by AAV-mediated gene transfer and found that TTR overexpression induced depression-like behavior (Figure 3B,C). These results suggest that chronic stress upregulates Ttr expression in the hippocampus, and that the produced TTR protein may be involved in the induction of depression-like behavior. In fact, several studies have reported an association between Ttr expression level and depression [30,36,37]. Sullivan et al. reported that TTR levels in the CSF of patients with depression were lower than in healthy controls [30]. Frye et al. have reported that TTR levels in the serum of bipolar depressed patients exceeded those in healthy controls [36]. Sousa et al. showed that the absence of TTR reduced the signs of depression-like behavior in Ttr-knockout mice [37], corroborating our results that TTR overexpression induces depression-like behavior. However, Stankiewicz et al. have reported that TTR gene expression was not altered in the hippocampus of mice with chronic social stress [15].

Furthermore, we found that TTR overexpression in the hippocampus induced the expression of Lcn2, Icam1, and Vcaml (Figure 3E–G), and these genes were also significantly increased in susceptible R-SDS mice but not in resilient R-SDS mice (Figure 3H–J). Lcn2 exhibited the highest level of upregulation in the susceptible R-SDS mice (Figure 2J), indicating that its expression was stimulated by excess TTR but not directly by chronic stress. LCN2 is a neutrophil gelatinase-associated protein that influences the immune system, iron homeostasis, lipid metabolism, and inflammatory responses [38]. In nonalcoholic steatohepatitis, elevated Lcn2 induces neuroinflammation through the release of HMGB1, resulting in blood–brain barrier dysfunction [39]. Stankiewicz et al. reported that Lcn2 expression was increased in the hippocampus of mice with chronic social stress [15], corroborating our results. In contrast, Lcn2 can act as a protective factor in the central nervous system in response to systemic inflammation [40]. Icam1 and Vcam1 are adhesion molecules on vascular endothelial cells that are essential for regulating leukocyte trafficking from blood vessels into tissues during inflammatory responses [41,42], and Lcn2 can regulate the migration and adhesion of neutrophils [43]. Sawicki et al. showed that social defeat stress increased Icam1 and Vcam1 expression in the vasculature of mouse brain regions, which are implicated in fear and anxiety responses [44]. Abcouwer et al. also reported that retinal ischemia-reperfusion injury, a model of retinal neurodegeneration with a breakdown of the blood–retinal barrier, induced the expression of inflammatory genes such as Lcn2 and Icam1, which led to leukocyte adhesion and vascular permeability, resulting in retinal neuroinflammation [45].

Thus, the findings in this study may suggest a possible mechanism of stress-induced depression in which chronic stress upregulates the expression of TTR and TTR stimulates the expression of inflammation-related genes such as Lcn2, Icam1, and Vcam1, which in turn cause inflammatory responses, ultimately resulting in depression-like behavior coupled with cytotoxicity by TTR. However, there is a consideration regarding the current study. The limitation of this study is that our findings do not necessarily apply to both male and female mice models because we only looked at male mice. Sex differences remain unknown and should be investigated further.

## 4. Materials and Methods

### 4.1. Animals

Male C57BL/6JJmsSlc (B6) and male Slc:ICR (ICR) mice retired from breeding were purchased from Japan SLC (Shizuoka, Japan). All mice were maintained at a temperature of 23 °C under constant 12 h light/dark cycles (light period from 7:00 a.m. to 7:00 p.m.) with food and water available ad libitum at Kanazawa Medical University. The Safe Harbor Mouse Retreat (The Jackson Laboratory Japan, Yokohama, Japan) was used for an enriched environment for the mice. All procedures for animal use and care were in accordance with the animal protocols of Kanazawa Medical University (Protocol Numbers 2017-117, 2020-42).

### 4.2. Forced Swim Stress (FSS)

This procedure was performed as described by Sun et al. [20] from 9:00 a.m. to 11:00 a.m. Nine-week-old male B6 mice were used in this experiment. Male B6 mice were placed in an acrylic cylinder containing water at 25 °C and at a depth of 25 cm so that they could neither escape nor touch the bottom (Figure 1A). These mice were forced to swim for 11 min daily for 5 consecutive days. After 1 min latency time, the behavior (moving distance and immobility) was recorded and measured for 10 min using ANY-maze video tracking software (version 6.35; Stoelting Co., Wood Dale, IL, USA).

### 4.3. Repeated Social Defeat Stress (R-SDS)

This procedure was performed as described by Goto et al. [46] with minor modifications from 1:00 p.m. to 4:00 p.m. [47]. Before R-SDS, male ICR mice were screened for their aggressiveness against a novel male B6 mouse for 3 min daily for 3 days. We evaluated the aggressiveness of the ICR mice based on the latency time and the number of attacks during the observation period, and we used only mice whose aggressiveness was stable. Before R-SDS, male B6 mice were individually housed with free access to food and water for one week. White-color partitioning boards were placed between every single cage so that the mice were not affected by the behaviors of neighboring mice. Three days before R-SDS, ICR mice were moved into a compartment of each cage (width × depth × height = 213 mm × 320 mm × 130 mm), which was divided by an acrylic divider containing holes to allow the mice to establish their territories in the cage (Figure 2A). A nine-week-old male B6 mouse, after isolation, was transferred to the territories of a male ICR mouse in the cage for 10 min daily for 10 days. After the physical contact time, a male B6 mouse was placed into another compartment next to the male ICR mouse in the cage until exposure to physical stress the next day so that a B6 mouse was exposed to various emotional stresses, including visual, auditory, and olfactory stimuli, from an ICR mouse for 24 h every day. The pairs of male B6 mice and male ICR mice were randomized daily to minimize variability in the aggressiveness of male ICR mice. Nonstressed control B6 mice were placed into each compartment divided by the divider to keep the mice in pairs in the cage and exchanged the pair every day.

### 4.4. Tail Suspension Test (TST)

This test was conducted as described by Cryan et al. [48] from 9:00 a.m. to 11:00 a.m. [47]. Male B6 mice were suspended from a rod of Bioseb’s Tail Suspension Cages by their tails using tape. Following failed attempts to escape from this situation, they are supposed to experience a despair-like state and become immobile. After 1 min latency time for taking motions, the movements of the mice were measured for 7 min by Bioseb’s Tail Suspension Test system (BIOSEB, Vitrolles, France). We evaluated immobile time; immobility was defined as the absence of an escaped-oriented movement. A significant increase in immobile time was considered a depression-like behavior.

### 4.5. Social Interaction Test (SIT)

This test was performed as described by Goto et al. [46] and Higashida et al. [49] with minor modifications from 10:00 a.m. to 2:00 p.m. [47]. After R-SDS for 10 days, R-SDS-induced depression model mice (R-SDS mice) and control mice were tested for their social interactions to identify subgroups of mice that were susceptible and resilient to social defeat stress. The B6 mice were placed in an open field chamber where a novel male ICR mouse (target) was enclosed in a metal meshwork at one end (Figure 2B). B6 mice were allowed to freely explore the chamber for 150 s under video recording and habituated to the same chamber in the absence of an ICR mouse for 150 s before SIT. The chamber was divided into an interaction zone (closest to the target), an avoidance zone (farthest from the target), and the other zone (Figure 2B). The time that each mouse spent in each zone was analyzed using SMART video tracking software (version 3.0; PanLab Harvard Apparatus, Holliston, MA, USA). The “avoidance zone score” was defined as the ratio of the time spent in the avoidance zone with an ICR mouse to the time spent in this zone without an ICR mouse [ICR(+)/ICR(−) ratio]. The “interaction zone score” was defined as the ratio of the time spent in the interaction zone with an ICR mouse to the time spent in this zone without an ICR mouse [ICR(+)/ICR(−) ratio]. Based on the avoidance zone score, the R-SDS mice with scores exceeding 1 were defined as “susceptible” to social defeat stress and those with scores less than 1 were defined as “resilient” to social defeat stress [23].

### 4.6. Sucrose Preference Test (SPT)

This test was conducted as described by Jiang et al. [14] with some modifications [47]. Two water bottles were presented to B6 mice for a 16 h habituation period. Next, two preweighted bottles (one containing water and the other containing 1% (*w/v*) sucrose solution) were presented to each mouse for 48 h. The positions of the water and sucrose bottles were switched every half day to avoid side bias. Bottles containing water and sucrose were weighed at the end of the study. The sum of the water and sucrose intakes was defined as the total intake, and sucrose preference was expressed as the percentage of sucrose intake relative to the total intake. A significant decrease in sucrose preference was considered a depression-like behavior.

### 4.7. RNA Isolation and Microarray

Fresh brains were immediately removed from euthanized animals and were kept in ice-cold PBS for 10 min. The hippocampus tissues dissected from the brain were homogenized in TRI Reagent (Molecular Research Center, Inc., Cincinnati, OH, USA), and total RNA was purified using an RNeasy Mini Kit (Qiagen, Hilden, Germany). RNA quality was assessed according to 28S/18S ratios of rRNA bands on electrophoresis gels under denaturing conditions. The microarray analysis was performed as previously described [50]. Briefly, 100 ng of total RNA was labeled according to the manufacturer’s instructions for the Ambion^®^ WT Expression Kit (Affymetrix, Santa Clara, CA, USA). Fragmented and labeled cDNAs were then hybridized onto the Affymetrix GeneChip Mouse Gene 1.0 ST arrays (Affymetrix). Arrays were washed and stained using the GeneChip Fluidics Station 450 and detected using a 3000 7G GeneChip Scanner (Affymetrix). All arrays passed the quality control criteria of the Expression Console software (version 4.0.3.14; Affymetrix). Raw data CEL files were then normalized using the RMA algorithm, and the data were exported using Expression Console. Three of the four mice analyzed by the microarray in the FSS were randomly selected and used for depicting the volcano plot. In the R-SDS, two mice were analyzed and used for the volcano plot. Genes with the absolute value of Log_2_(Fold-Change) > 0.5 and *p* < 0.05 were selected.

### 4.8. Quantitative Real-Time Reverse Transcription Polymerase Chain Reaction (RT-qPCR) Analysis

cDNA synthesis was conducted using purified RNA from the hippocampus of male B6 mice and a PrimeScript^TM^ RT Master Mix (TaKaRa, Otsu, Japan). Real-time PCR was performed using a StepOnePlus Real-Time PCR System (Applied Biosystems, Foster City, CA, USA) with TB Green^®^ Premix Ex Taq^TM^ II (TaKaRa) and specific primer pairs (Table 1), according to the manufacturer’s protocol. The PCR conditions were 30 s at 95 °C, followed by 40 cycles of 5 s at 95 °C and 30 s at 60 °C. *Gapdh* was used to normalize the relative expression levels of each target mRNA. All measurements were performed in duplicate, and the relative amount of detected target genes was calculated using the ΔΔCt method.

### 4.9. Western Blotting

Equal amounts of protein samples were subjected to SDS-PAGE on a 12.5% polyacrylamide gel (FUJIFILM Wako Chemicals, Osaka, Japan). After electrophoresis, the proteins were transferred onto PVDF membranes (Merck Millipore, Burlington, MA, USA) and blocked with 5% skim milk in TBST (20 mM Tris, 150 mM NaCl, 50 mM KCl, and 0.05% Tween 20). Anti-TTR (1:500, Cat. No. ab215202, Abcam, Cambridge, UK), anti-LCN2 (1:1000, Cat. No. 26991-1-AP, Proteintech Group Inc., Rosemont, IL, USA), anti-ICAM1 (1:1000, Cat. No. ab179707, Abcam), anti-VCAM1 (1:1000, Cat. No. ab134047, Abcam), and anti-ACTB (1:2000, Cat. No. MA5-15739, Sigma-Aldrich, St. Louis, MO, USA) antibodies in blocking buffer were used as the secondary antibodies. Protein signals were detected using a Western Blot Hyper HRP Substrate (TaKaRa).

### 4.10. Adeno-Associated Virus (AAV)-Mediated Gene Transfer

cDNA synthesis from the purified hippocampal RNA was performed using a PrimeScript^TM^ RT Master Mix (TaKaRa). The coding region of mouse Ttr (NM_013697.5) was amplified with primers listed in Table 1 and cloned by insertion of Ttr between *Eco*R I and *Bam*H I sites of the pAAV-CMV vector (pAAV-TTR) using In-Fusion HD^®^ Cloning Kit (TaKaRa). pAAV-TTR was cotransfected with the plasmids pRC2-mi342 vector and pHelper vector in HEK293T cells using the AAVpro Helper Free System (AAV2) (TaKaRa) and calcium phosphate method. The pAAV-ZsGreen1 vector (TaKaRa) was cotransfected in the same manner as the control. After 48 h, the cells were harvested, and the crude AAV vector solution was purified and concentrated using the AAVpro^®^ Purification Kit (TaKaRa). The titers of AAV-TTR and AAV-ZsGreen1 were measured to be 2.5 × 10^11^ Vg/mL and 2 × 10^11^ Vg/mL, respectively. AAV vectors were bilaterally injected at 2 × 10^8^ Vg into the cornu ammonis area (−2.2 mm anteroposterior, ±1.8 mm mediolateral, −1.5 mm dorsoventral from the bregma) (Appendix A). AAV experiments were approved by the Kanazawa Medical University Committee for Recombinant DNA Experiment (Protocol Number: 2015-7, 2020-9).

### 4.11. Statistical Analysis

Statistical significance of the data was determined using Student’s t-test for unpaired data with equal variances and the Mann–Whitney U-test for unpaired data with unequal variances. Multiple comparisons were analyzed by one-way ANOVA Tukey–Kramer test using the StatPlus:mac LE (version 8; https://www.analystsoft.com/en/ (accessed on 15 May 2021)).

## 5. Conclusions

In this study, we prepared two depression mouse models, FSS mice and R-SDS mice, and examined gene expressions in the hippocampus of these mice. Our study revealed that Ttr expression was commonly elevated in the hippocampus of these mice, and hippocampal TTR overexpression induced depression-like behavior. AAV-mediated TTR overexpression upregulated the expression of Lcn2, a modulator of inflammation, and other proinflammatory genes, such as Icam1 and Vcam1, in the hippocampus. The *Ttr* mRNA level was similarly increased in the hippocampus of susceptible R-SDS mice compared with the control, but no significant difference was observed between the hippocampus of the control and resilient R-SDS mice. These results thus suggest that chronic stress may induce the upregulation of Ttr and excess amounts of TTR to stimulate the expressions of proinflammatory genes, which cause inflammatory responses, resulting in the development of depression-like behavior coupled with cytotoxicity by TTR.

## Figures and Tables

**Figure 1 ijms-24-03736-f001:**
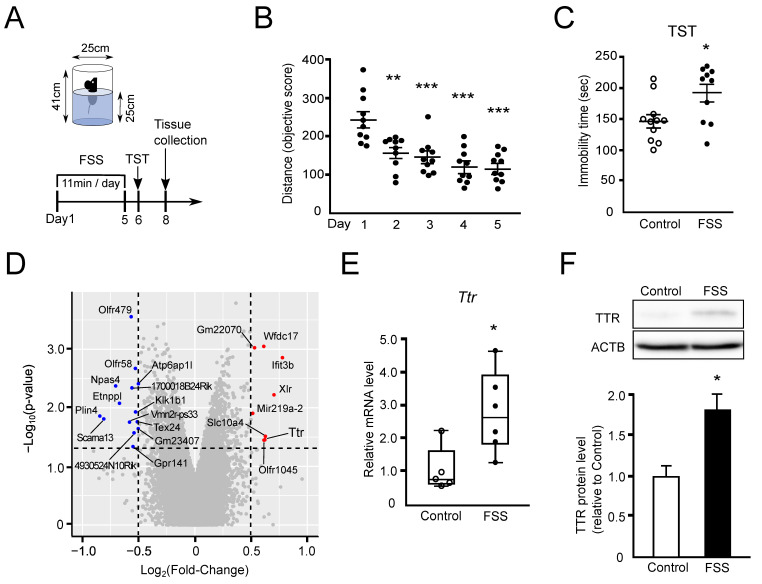
Preparation of forced swim stress (FSS)-induced depression mouse model, behavioral tests, and gene expression analyses. (**A**) Schedule of mouse preparation and analyses. Male B6 mice were loaded with FSS in an acrylic cylinder for 10 min daily for 5 consecutive days. After FSS loading, the mice received the tail suspension test (TST) on Day 6. Then, the hippocampus from male B6 mice was collected on Day 8 for gene expression analyses. (**B**) Moving distance of the mice during FSS loading. Values of each individual were plotted, and mean ± SEM per group was also expressed. ** *p* < 0.01, *** *p* < 0.001 vs. Day 1. n = 10. (**C**) The immobile time in TST on Day 6. For the control, the time was measured in the nonstressed mice. Values of each individual were plotted, and mean ± SEM per group was also expressed. Control; n = 11, FSS; n = 10. (**D**) Volcano plot representing gene expression changes in the hippocampus of the FSS mice. The fold-change and the significance are converted to Log_2_(Fold-Change) and −Log_10_(*p*-value), respectively. The vertical and horizontal dotted lines show the cut-off of Log_2_(Fold-Change) = ± 0.5 and of *p* = 0.05, respectively. (**E**) Ttr mRNA expression in the hippocampus of FSS mice was quantified by RT-qPCR (n = 5–6). The box represents the 25–75th percentiles, and the median is indicated. The whiskers represent the highest and lowest values. (**F**) TTR protein expression in the hippocampus of FSS mice was analyzed by Western blotting. Data are expressed as mean ± SEM (n = 3). * *p* < 0.05.

**Figure 2 ijms-24-03736-f002:**
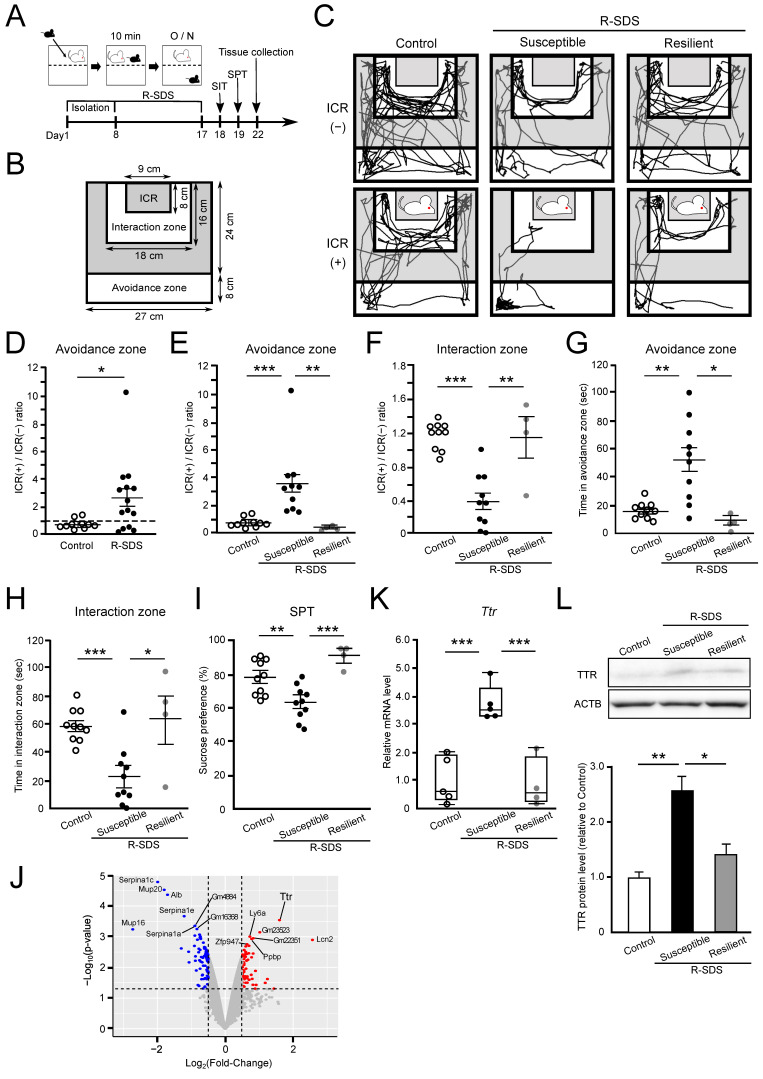
Preparation of repeated social defeat stress (R-SDS)-induced depression mouse model, behavioral tests, and gene expression analyses. (**A**) Schedule of mouse preparation and analyses. After social isolation for a week, male B6 mice were subjected to R-SDS for 10 min daily for 10 consecutive days, from Day 8 to Day 17. The mice received the social interaction test (SIT) on Day 18 and the sucrose preference test (SPT) on Day 19. Then, the hippocampus from male B6 mice was collected on Day 22 for gene expression analyses. (**B**) The apparatus used for SIT. The apparatus was separated into 4 areas: the ICR area, the interaction zone, the avoidance zone, and the intermediate zone between the interaction and avoidance zones. The ICR area was separated from the other 3 areas by using a metal meshwork. There were no obstacles between the other 3 areas. In the SIT, a social target ICR mouse was placed in the ICR area and an R-SDS mouse or control mouse receiving SIT was placed in the other 3 areas, and the time that the mouse spent in the interaction zone and the time in the avoidance zone were measured. For the control, the time was measured as described above, but in the absence of a social target ICR mouse. (**C**) Typical traces of movement of the control, susceptible R-SDS, and resilient R-SDS mice in the presence [ICR(+)] or absence [ICR(−)] of ICR mice. (**D**) Avoidance zone score [ICR(+)/ICR(−) ratio] of the control (n = 10) and whole R-SDS (n =14) mice group. Dashed line represented 1 of the avoidance zone score. (**E**,**F**) Avoidance zone (**E**) and interaction zone (**F**) scores [ICR(+)/ICR(−) ratio] of the control, susceptible R-SDS, and resilient R-SDS mice. (**G**,**H**) Total time spent in the avoidance zone (**G**) and interaction zone (**H**) of the control, susceptible R-SDS, and resilient R-SDS mice. (**I**) Sucrose preference of the control, susceptible R-SDS, and resilient R-SDS mice in SPT. (**E**–**I**) Control; n = 10, susceptible; n = 10, and resilient; n = 4. (**D**–**I**) Values of each individual were plotted, and mean ± SEM per group was also expressed. (**J**) Volcano plot representing gene expression changes in the hippocampus of the susceptible R-SDS mice. The fold-change and the significance are converted to Log_2_(Fold-Change) and −Log_10_(*p*-value), respectively. The vertical and horizontal dotted lines show the cut-off of Log_2_(Fold-Change) = ± 0.5 and of *p* = 0.05, respectively. (**K**) Ttr mRNA expressions in the hippocampus of the control, susceptible R-SDS, and resilient R-SDS mice were quantified by RT-qPCR (n = 4–5). The box represents the 25–75th percentiles, and the median is indicated. The whiskers represent the highest and lowest values. (**L**) TTR protein expressions in the hippocampus of the control, susceptible R-SDS, and resilient R-SDS mice were analyzed by Western blotting. Data are expressed as mean ± SEM (n = 4). * *p* < 0.05, ** *p* < 0.01, *** *p* < 0.001.

**Figure 3 ijms-24-03736-f003:**
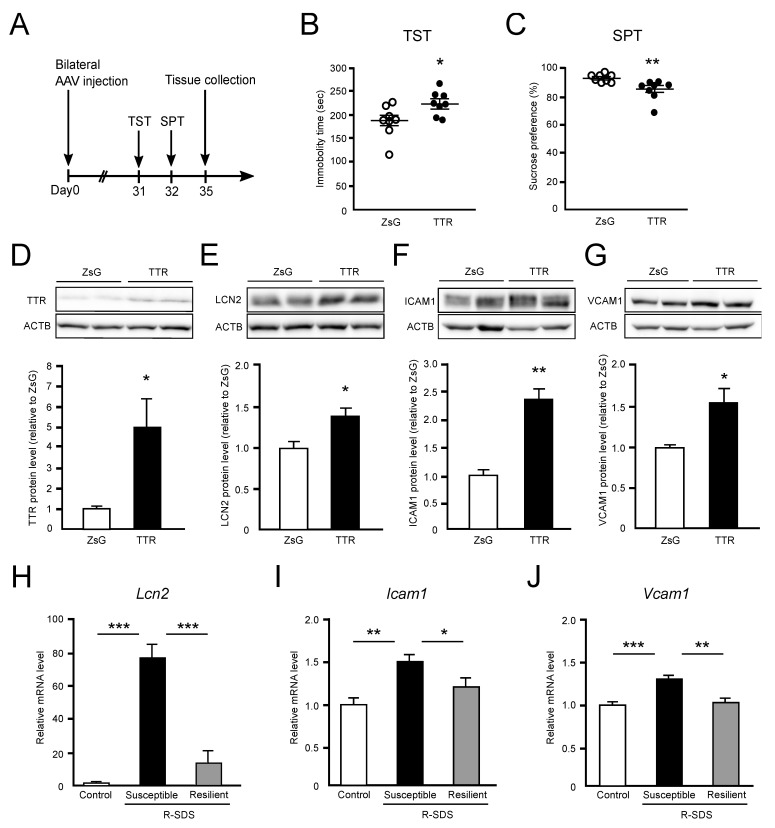
Effects of Ttr overexpression in the hippocampus. (**A**) Schedule of gene transfer and mouse analyses. Male B6 mice were injected bilaterally with AAV-ZsGreen1 (ZsG mice) or AAV-Ttr (TTR mice) into their hippocampus. TST, SPT, and tissue collection were performed 31, 32, and 35 days after AAV injection, respectively. (**B**,**C**) The immobile time in TST (**B**) and sucrose preference (**C**) of ZsG and TTR mice. Values of each individual were plotted, and mean ± SEM per group was also expressed. ZsG; n = 8, TTR; n = 8. (**D**–**G**) The protein levels of TTR (**D**), LCN2 (**E**), ICAM1 (**F**), and VCAM1 (**G**) in the hippocampus of the ZsG and TTR mice. Western blotting was performed using whole hippocampus extracts of the ZsG and TTR mice. Data were expressed as mean ± SEM (n = 4). (**H**–**J**) The mRNA levels of *Lcn2* (**H**), *Icam1* (**I**), and *Vcam1* (**J**) in the hippocampus of the control, susceptible R-SDS, and resilient R-SDS mice. The mRNA levels were analyzed by RT-qPCR with RNA extracted from the hippocampus of the control, susceptible R-SDS, and resilient R-SDS mice. Data were expressed as mean ± SEM (n = 4–5). * *p* < 0.05, ** *p* < 0.01, *** *p* < 0.001.

**Table 1 ijms-24-03736-t001:** Primers for RT-qPCR and oligonucleotides for AAV-TTR.

Gene	Forward Primer	Reverse Primer
*Ttr*	5′-CGCGGATGTGGTTTTCACAG-3′	5′-AATTCTGGGGGTTGCTGACG-3′
*Lcn2*	5′-ATGTCACCTCCATCCTGGTCAG-3′	5′-GCCACTTGCACATTGTAGCTCTG-3′
*Icam1*	5′-TTTGAGCTGAGCGAGATCGG-3′	5′-CGGAAACGAATACACGGTGATG-3′
*Vcam1*	5′-GCTATGAGGATGGAAGACTCTGG-3′	5′-ACTTGTGCAGCCACCTGAGATC-3′
*Gapdh*	5′-TGACGTGCCGCCTGGAGAAAC-3′	5′-CCGGCATCGAAGGTGGAAGAG-3′
AAV-TTR	5′-GGATTCGCGAGAATTATGGCTTCCC TTCGACTCTTCC-3′	5′-TGCCACCCGTGGATCTCAATTCTGG GGGTTGCTGAC-3′

## Data Availability

The DNA microarray data presented in this study are available in the Gene Expression Omnibus (GEO) under SuperSeries Accession Number GSE218809. This SuperSeries comprises SubSeries Accession Numbers GSE223430 (Figure 1D) and GSE218742 (Figure 2J).

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
