# Peer review of "Transthyretin Is Commonly Upregulated in the Hippocampus of Two Stress-Induced Depression Mouse Models"

_ijms, 2023, doi:10.3390/ijms24043736_

Round 1

Reviewer 1 Report

In this manuscript, Saito-Takatsuji et al sought to profile gene expression changes in the hippocampus under stressed conditions in mouse models of human major depression. As models, the authors used both the forced swimmming stress and the chronic social defeat. Using only male mice, the authors found only one robust genetic dysregulation common to both stress models and involving transthyretin (Ttr). To demonstrate the functional role of Ttr upregulation in depression-like behaviour, the authors overexpressed it in the hippocampus by AAV-mediated gene transfer. This resulted after 35 days in increased despair and anhedonia. Finally, three other proinflammatory genes showed variations of expression consequent to Ttr-induced stress susceptibility.

In the absence of a robust biomarker of major depression, the work is of interest and is well written, with most conclusions supported by the results presented. Nevertheless, we see some limitations in the interpretation of the results.

Major concerns:

-For the first set of experiments, it is not known how many samples were analyzed on microarrays. Four independent experiments are illustrated but it is not explained in the methods how these independent experiments were conducted or the number of samples used for each experiment. Why were all the samples not processed in one run but in 4 runs? The exact same question applies for the second set of experiments (2 independent experiments, Fig.2J). We would be curious to get an idea of what would happen if you present the results as volcano plots to have an idea of the distribution of all expressed genes between control and stressed animals for forced swimming and chronic social defeat. In this case, we wonder if you could find other dysregulated genes in both stress models with a corrected p-value <0.05.

-We do not doubt that authors have experience in conducting AAV-mediated gene transfer but they should provide proofs that stereotaxic injections targeted the right area. They should also provide and approximate quantification of local Ttr mRNA overexpression compared to control injection.

-In Figure 3H, it appears that Lcn2 is upregulated nearly 80 times in susceptible mice subjected to chronic social defeat comparted to controls: this is a huge variation in expression. Are you sure there is not an error in the mRNA quantification?

Minor concerns:

-The introduction is concise and straight to the point but we feel that the authors have overlooked some important work done by other investigators and it is important to cite them: Bagot et al PMID 27181059, Gammie PMID 34997033 and von Ziegler et al PMID 35383160. It would be important for the authors to carefully review these studies and explore in additional data whether transthyretin is also commonly dysregulated by stress as an indicator of depression. In addition, for discussion, the authors may have missed the work of Turner et al PMID 24795602 who pointed out that transthyretin is the most highly expressed known transcript in choroid plexus

-The present work involves only male animals. We suggest that authors consider adding limitation to their conclusions and discuss the sex issue, important in the depression pathophysiology

-In Materials and Methods, for Western blotting, could you precise what is the concentration (or dilution) of the used antibodies.  

-In Figure1E and 2K, instead of histograms, please display individual data as dot plot and Tukey box.

Reviewer 2 Report

The manuscript entitled “Transthyretin is commonly up-regulated in the hippocampus of two stress-induced depression mouse models” describes chronic stress related induction of transthyretin in the hippocampus by forced swim and social defeat stress in C57Bl/6 mice. Overall, results are clear and conclusions are reasonable.

In Figure 1C, there are 2 mice in the control group showing higher immobility time (about 200 sec) than others (100 to 150 sec). Would there be difference in gene expression in those 2 mice? Also, there are 3 mice showing less immobility in the FSS group. Would there be a difference in Ttr expression?

There are several papers showing that CSF transthyretin decreases in the CSF of depression patients. Can authors discuss about this discrepancy?

(Sullivan et al. 1999 https://doi.org/10.1176/ajp.156.5.710)

(DOI: 10.1176/ajp.150.5.813)

(https://doi.org/10.1016/j.neulet.2009.06.025)

Round 2

Reviewer 1 Report

We thank the authors for adressing absolutely all the concerns that we raised and for significantly improving their manuscript.

Author Response

We really appreciate for your valuable comments for improving our study.